# Investigation of the Polymer Coextrusion Process: A Review

**DOI:** 10.3390/polym14071309

**Published:** 2022-03-24

**Authors:** Jean-François Agassant, Yves Demay

**Affiliations:** 1MINES Paris, PSL Research University, CEMEF, UMR CNRS 7635, Sophia-Antipolis, 06560 Valbonne, France; 2Laboratory J.A. Dieudonné, UMR CNRS 7351, University Côte d’Azur, Parc Valrose, 06000 Nice, France; yves.demay@unice.fr

**Keywords:** polymer coextrusion, thickness homogeneity, instabilities, encapsulation

## Abstract

A review of the different coextrusion processes and the related processing problems is presented. A one-dimensional bilayer coextrusion Poiseuille flow model is first developed with Newtonian and shear-thinning rheological behaviors. A transitory computation at the convergence between the two independent polymer layers shows that stationary interface position and velocity profile are established after a short distance of the order of the die gap which justifies the validity of the 1D stationary model. This model is then applied to multilayer temperature dependent coextrusion flows which correspond to realistic industrial coextrusion conditions. Marked interface instabilities may be observed depending on the rheology of the coextruded polymers and of their flow rate ratios. Experiments point clearly out that these instabilities may be amplified along the die land. Convective stability analysis as well as direct numerical computation discriminate flow situations which amplify or damp down instabilities. These 1D models are unable to account for the complex feedblock coat-hanger die geometries. A thin layer coextrusion model is then developed, based on the Hele-Shaw lubrication approximations already used for single layer extrusion problems. It allows to predict the location of the interfaces between the different layers in the whole die, and especially at die exit. This represents a major issue in feedblock die coextrusion. These thin layer approaches are unable to address the encapsulation of one polymer by the other in these complex die geometries with important gap thicknesses. Experiments conducted in dies of square section allow identifying the dynamics of encapsulation. 3D models are required to account for this phenomenon but the management of the sticking contact at the die wall poses difficult numerical problems.

## 1. The Coextrusion Processes: Interest, Technology and Limiting Problems

The coextrusion process consists in the simultaneous flows within the same equipment of different polymers with various characteristics in order to obtain a multilayer product having particular properties. The idea is to combine in a single product the properties of the constituent polymers (impact and scratch resistance, appearance, barrier properties), while achieving a good adhesion between the different layers. In some cases, it may allow to use a recycled polymer between two layers of the same virgin polymer, thus decreasing the price of the product. These processes have in recent years experienced a strong development in the field of packaging, coextrusion having particularly widespread throughout the food industry. From initial configurations that combined two products in two or three layers, it is now common to produce complex configurations involving up to 5 or 6 different polymers in structures that can be composed by up to ten layers [1] For a typical application in food bottles, one can find for example an inner polypropylene layer, a tie layer for adhesion between incompatible layers, a barrier layer to oxygen (e.g., ethyl vinyl alcohol, EVOH), a second binder layer, a layer of recycled polypropylene to reduce the material cost and a final layer of virgin polypropylene to ensure a good surface aspect.

Obviously, these coextruded products now pose environmental problems due to the difficulty of dissociating the constituent polymers at the end of their life for recycling, but their properties cannot be obtained, so far, with a single polymer. In food packaging for example, where coextruded products are widely used, they preserve food quality and thus contribute to the improvement of human nutrition. On the other hand, coextrusion is a clever way to use important quantities of recycled polymers for the inner layers of pipes, sheathings of cables and profiles.

In the first section of this paper we review the different coextrusion processes and the related problems. In the second section we present the 1D model in situations of increasing complexity: Newtonian or shear-thinning behaviors, bilayer or multilayer coextrusion flows, isothermal and non-isothermal. These 1D models are then used to understand the occurrence of coextrusion instabilities. The third section is devoted to the more complex feedblock coat-hanger die situation. A generalized thin layer Hele-Shaw computation allows predicting the polymer flow distribution in each layer of the final coextruded film or sheet. The forth section is devoted to the encapsulation problem encountered in thick coextrusion die geometries which remains not totally solved at present. The review is illustrated by several experimental results.

### 1.1. The Different Coextrusion Processes

Extrusion processes concerned by multilayer flows are blow-molding for the production of hollow bodies (bottles, tanks, etc.), film blowing, flat dies for the manufacture of sheets for thermoforming, pipe, cable coating and profile dies.

Figure 1 presents an example of geometry in the case of a tri-layer film blowing die. In the same tool are stacked as many helical dies (A, B, C) as different polymers have to be coextruded. The truly multilayer flow occurs here in the terminal area of the lips on a short distance. This leads to very complex and very expensive tools but, as each die is optimized for the corresponding polymer, a good distribution of the various layers’ thicknesses is obtained.

During the last ten years, a new type of coextrusion die has been specially developed for film production, called pancake die. As shown in Figure 2, this die follows the classic principle of flow in helical channels, except that they are laid flat and that polymer now flows from the periphery to the center. The major advantage of this design is for coextrusion of multilayer films, where the number of layers of different polymers may be multiplied by stacking pancake dies on top of each other. It is possible now with this technique to produce blown films containing up to nine different layers.

The same kind of disposal is used for the production of multilayers hollow bodies.

Two different systems can be used in coextrusion of films or plates. First, the multichannel die (Figure 3).

In the same tool, there are various coat-hanger sections (A, B, C) to distribute the different polymers and a common flow section (L) at the lips. It provides an excellent distribution of the polymers, but combining the different coat-hangers lead to a complex and expensive die geometry. Moreover, its design limits for reasons of space the number of layers of different polymers that can be coextruded. However, it allows the use of highly different polymers (e.g., in viscosity), since the actual coextrusion flow is carried out only on a short distance before the outlet of the die.

The second system is much more common and makes use of a conventional flat die preceded by a feedblock, wherein different polymers are brought into contact before entering the die (Figure 4). This feedblock can be more or less sophisticated [3]. It generally consists of a zone with different lamellas (zone D in Figure 4) for which the geometry can be changed in order to feed the flat die with a stratified structure, which is well controlled. The different polymers flow then conveniently in the coat-hanger section (zone P) and through the lips (L). This second system is much cheaper and much more flexible than a multichannel die. It is often preferred even if the distribution problems are a bit more difficult to control.

The most common devices for tube or cable coating coextrusion dies correspond to “coat hanger” type dies which are somehow wound (Figure 5) around the die mandrel in which the polymer of the tube inner layer is flowing. As for the film blowing dies (Figure 1 and Figure 2) and for the multi-channel sheet dies (Figure 3), it is necessary to control the distribution of the polymer in each “coat-hanger” in order to obtain a coextruded tube with a uniform thickness for each layer. It can be seen in Figure 5 that the meeting point between the two external polymers (blue and red) precedes the meeting point with the internal layer of the tube.

### 1.2. Problems Encountered

The major problems that are encountered in coextrusion are of the same type as those for single material flows: control of material distributions at the die exit, uniformity of thickness and temperature of each layer, etc. The focus will be made on these distribution problems in this review paper.

However, special phenomena related to the multiphase nature of the flow will appear:

Lack of adhesion between the layers: it is often necessary to add a thin layer of glue (or tie-layer) between two incompatible polymers. A uniform distribution of this very thin tie-layer (a few micrometers) between two adjacent thick incompatible polymer layers is mandatory.

Interfacial instabilities that may occur at the interface between two adjacent layers of polymers, whereas the whole extrudate is free of extrusion defects. These defects are particularly apparent when manufacturing coextruded films that involves very thin tie-layers between the main polymer streams. They appear in the form of waves on all or parts of the width of the sheet as illustrated in Figure 6.

When two coextruded polymers flow in a long die, a progressive encapsulation phenomenon may be observed (Figure 7). When dealing with two Newtonian fluids of different viscosities, the only thermodynamically stable configuration is one in which the less viscous fluid fully encapsulates the more viscous one. When dealing with non-Newtonian shear-thinning polymer flows, this is less obvious as the viscosity ratio will change along the flow depending on the local shear rates and temperatures.

### 1.3. Coextrusion Modeling: A Tool for Understanding the Process

Polymer coextrusion flow modeling provides information about the velocity field, the shear rate, stress and temperature distributions.

In a first step we study the Poiseuille coextrusion flow of Newtonian and shear-thinning viscous fluids. It will be applied to a realistic multilayer coextrusion situation and then to the process stability analysis.

We then develop Hele-Shaw thin layer computations in order to study the coextrusion flow in a coat-hanger die and finally we investigate the complex encapsulation phenomenon.

## 2. Poiseuille Coextrusion Flow

This corresponds to the coextrusion flow in the final land of film blowing dies, multichannel sheet dies, tube dies. Starting from the academic situation of the isothermal coextrusion of two Newtonian fluids between parallel plates, more complex situations will be progressively addressed: Coextrusion of two shear-thinning fluids, non-isothermal multilayer coextrusion, transitory phenomena at the coextrusion die inlet.

Early computations have been performed in isothermal conditions for three layers sandwich flow between parallel plates [6] and for annular flow ([7,8]) using a power law rheology. A Carreau law has been used in [9] which allows accounting for the transition between a Newtonian behavior at low shear rates and a shear thinning behavior at high shear rates. Non-isothermal effects have been introduced in ([10,11,12,13]). In this section, the computations proposed by [2,14,15] will be developed. These 1D computations are well suited to flow predictions in the terminal land of film blowing dies (Figure 1 and Figure 2), multi-channel sheet dies (Figure 3) or tube dies (Figure 5) which geometries are very near annular or parallel plates flow configurations. They represent a prerequisite for the stability analysis of the coextrusion flow which is the major problem especially when using polymers with very different rheological behaviors or when introducing very thin tie-layers.

### 2.1. Stationary Isothermal Coextrusion Flow of Two Newtonian Fluids

Consider the flow situation depicted in Figure 8. Two fluids of viscosities $\eta_{1}$ and $\eta_{2}$ respectively, flow under the effect of a pressure drop Δ*p* between two parallel plates of gap *h*, length *L* and width *W*. The flow is uniform within the width which means that encapsulation phenomena are not accounted for. One calculates the flow rate of each layer as a function of the position of the interface and obtains the corresponding velocity profiles. The position of the interface between the two fluids, *αh* (0 ≤ *α* ≤ 1), is a priori unknown. To calculate the velocity field in each layer ($u_{i}=(u_{i}(z),0,0)$ for *i* = 1, 2) and the flow rate-pressure relationships, we must solve the Stokes equations in each layer.
(1)$$\left\{ \begin{array}{l} \frac{dp_{i}}{dx}=\eta_{i}\frac{d^{2}u_{i}}{dz^{2}}\\ \frac{dp_{i}}{dz}=0 \end{array}\right.$$

The second equation implies that the pressure is independent of *z* and so identical in both layers at each abscissa *x* and:(2)$$\frac{dp_{i}}{dx}=-\frac{\Delta p}{L}$$

The velocity field is obtained by integration of Equation (1):(3)$$u_{i}(z)=-\frac{1}{2\eta_{i}}\frac{\Delta p}{L}z^{2}+C_{i}z+D_{i}$$

Imposing a sticky contact of fluid 1 at the lower plate implies *D*_1_ = 0.

The velocity is continuous at the interface between the two fluids: $u_{1}(\alpha h)=u_{2}(\alpha h)$.

The shear stress is also continuous: $\eta_{1}\frac{du_{1}}{dz}(\alpha h)=\eta_{2}\frac{du_{2}}{dz}(\alpha h)$.

There is also a sticky contact between fluid 2 and the upper plate: $u_{2}(h)=0$.

This results in a linear system of three equations with three unknowns *C*_1_, *C*_2_, *D*_2_:(4)$$\left\{ \begin{array}{l} -(\alpha h)C_{1}+(\alpha h)C_{2}+D_{2}=-\frac{1}{2\eta_{1}}\frac{\Delta p}{L}{\left(\alpha h\right)}^{2}+\frac{1}{2\eta_{2}}\frac{\Delta p}{L}{\left(\alpha h\right)}^{2}\\ -\eta_{1}C_{1}+\eta_{2}C_{2}=0\\ C_{2}h+D_{2}=\frac{1}{2\eta_{2}}\frac{\Delta p}{L}h^{2} \end{array}\right.$$

The velocity profile writes in each layer:(5)$$u_{1}(z)=\frac{\Delta p}{2\eta_{1}L}z(-z+h\frac{\alpha^{2}(\eta_{1}-\eta_{2})-\eta_{1}}{\alpha (\eta_{1}-\eta_{2})-\eta_{1}})$$
(6)$$u_{2}(z)=\frac{\Delta p}{2\eta_{2}L}(-z^{2}+zh\frac{\alpha^{2}(\eta_{1}-\eta_{2})-\eta_{1}}{\alpha (\eta_{1}-\eta_{2})-\eta_{1}}+h^{2}\frac{\alpha (1-\alpha )(\eta_{1}-\eta_{2})}{\alpha (\eta_{1}-\eta_{2})-\eta_{1}})$$

The flow rate in each layer is obtained by integration:(7)$$Q_{1}=W\int_{0}^{\alpha h}u_{1}(z)dz=W\frac{\Delta ph^{3}\alpha^{2}}{12\eta_{1}L}\cdot \frac{\alpha^{2}(\eta_{1}-\eta_{2})+2\alpha \eta_{1}-3\eta_{1}}{\alpha (\eta_{1}-\eta_{2})-\eta_{1}}$$
(8)$$Q_{2}=W\int_{\alpha h}^{h}u_{2}(z)dz=W\frac{\Delta ph^{3}{\left(1-\alpha \right)}^{2}}{12\eta_{2}L}\cdot \frac{\alpha (4-\alpha )(\eta_{1}-\eta_{2})-(2\alpha +1)\eta_{1}}{\alpha (\eta_{1}-\eta_{2})-\eta_{1}}$$

The ratio between both flow rates is only a function of the interface position α and of the viscosity ratio $\beta =\frac{\eta_{2}}{\eta_{1}}$
(9)$$\frac{Q_{1}}{Q_{2}}=\beta {\left(\frac{\alpha }{1-\alpha }\right)}^{2}\frac{\alpha^{2}(1-\beta )+2\alpha -3}{\alpha (4-\alpha )(1-\beta )-2\alpha -1}$$

Figure 9 reports the variations of the flow rates ratio as a function of the relative position of the interface *α* for different values of the viscosity ratio *β*. As *α* is an increasing function of the flow rates ratio for any value of *β*, the interface position is determined without ambiguity. For a given flow rates ratio the interface position increases with the viscosity ratio. This figure is very effective on an academic point of view but it is largely inappropriate for mastering coextrusion of molten polymers, because their viscosity depends on the shear rate which varies through the die gap and the viscosity ratio between both polymers has no real meaning.

### 2.2. Generalization to Shear-Thinning Polymer Fluids

In a first step we use a power law equation which allows to obtain analytical solution.
(10)$$\eta =K{\left|\dot{\gamma }\right|}^{m-1}$$

$\dot{\gamma }$ is the shear rate, *K* and *m* are the polymer consistency and the power law index. As previously, the Stokes equation in the *z* direction implies that the pressure is uniform through the gap at each abscissa *x*. The Stokes equation writes for each layer in the *x* direction:(11)$$\frac{dp}{dx}=\frac{d}{dz}\left[K{\left|\frac{du}{dz}\right|}^{m-1}\frac{du}{dz}\right]=-\frac{\Delta p}{L}$$

As in the Newtonian case, both velocity and shear stress are continuous at the interface:(12)$$u_{1}(\alpha h)=u_{2}(\alpha h)$$
(13)$$K_{1}{\left|\frac{du_{1}}{dz}\right|}^{m_{1}-1}\frac{du_{1}}{dz}=K_{2}{\left|\frac{du_{2}}{dz}\right|}^{m_{2}-1}\frac{du_{2}}{dz}$$

A zero velocity is imposed at the lower and the upper walls. The existence of an absolute value in Equations (11) and (13) makes the resolution more complex than previously because it is now necessary to identify the location of the maximum velocity *ch* which may be in layer 1 or layer 2 depending on their respective flow rates and rheologies. The shear rate $\dot{\gamma }$ is positive below *ch* and negative above *ch* and the Stokes equation has a different shape below and above *ch*.

The velocity profile writes in layer 1 which power law parameters are *K*_1_ and *m*_1_:(14)$$u_{1}(z)=\frac{m_{1}}{1+m_{1}}{\left(\frac{1}{K_{1}}\frac{\Delta p}{L}\right)}^{1/m_{1}}h^{(1+m_{1})/m_{1}}\left[c^{(1+m_{1})/m_{1}}-{\left|\frac{z}{h}-c\right|}^{(1+m_{1})/m_{1}}\right]$$
and in layer 2 which power law parameters are *K*_2_ and *m*_2_:(15)$$u_{2}(z)=\frac{m_{2}}{1+m_{2}}{\left(\frac{1}{K_{2}}\frac{\Delta p}{L}\right)}^{1/m_{2}}h^{(1+m_{2})/m_{2}}\left[{\left(1-c\right)}^{(1+m_{2})/m_{2}}-{\left|\frac{z}{h}-c\right|}^{(1+m_{2})/m_{2}}\right]$$

A relationship between the location of the interface αh and the location of the maximum velocity *ch* is deduced from Equations (12) and (13):(16)$$\frac{m_{1}(m_{2}+1)}{m_{2}(m_{1}+1)}h^{\left(1/m_{1}-1/m_{2}\right)}{\left(\frac{\Delta p}{L}\right)}^{\left(1/m_{1}-1/m_{2}\right)}\frac{K_{2}^{1/m_{2}}}{K_{1}^{1/m_{1}}}=\frac{{\left(1-c\right)}^{(1+m_{2})/m_{2}}-{\left|\alpha -c\right|}^{(1+m_{2})/m_{2}}}{c^{(1+m_{1})/m_{1}}-{\left|\alpha -c\right|}^{(1+m_{1})/m_{1}}}$$

Integrating Equations (14) and (15) leads to the flow rate in each layer:(17)$$Q_{1}=W\frac{m_{1}}{1+m_{{}^{1}}}{\left(\frac{1}{K_{1}}\frac{\Delta p}{L}\right)}^{1/m_{1}}h^{(2m_{1}+1)/m_{1}}\;\left[\frac{1+m_{1}}{1+2m_{1}}c^{(2m_{1}+1)/m_{1}}+(\alpha -c)c^{(1+m_{1})/m_{1}}-\frac{m_{1}}{2m_{1}+1}(\alpha -c){\left|\alpha -c\right|}^{(1+m_{1})/m}\right]$$



(18)
$$Q_{2}=W\frac{m_{2}}{1+m_{{}^{2}}}{\left(\frac{1}{K_{2}}\frac{\Delta p}{L}\right)}^{1/m_{2}}h^{(2m_{2}+1)/m_{2}}\;\left[\frac{1+m_{2}}{1+2m_{2}}{\left(1-c\right)}^{(2m_{2}+1)/m_{2}}+\left(c-\alpha \right){\left(1-c\right)}^{(1+m_{2})/m_{2}}-\frac{m_{2}}{2m_{2}+1}(c-\alpha ){\left|c-\alpha \right|}^{(1+m_{2})/m_{2}}\right]$$



The complexity of the results no longer makes it possible to define simple charts, of the type of those in Figure 9. It is interesting to illustrate these results by a representation of velocity profiles. Figure 10 shows two profiles corresponding to a flow rates ratio $Q_{1}/Q_{2}$ of 10.8 for two different consistency values of fluid 1 but the same consistency for fluid 2. It is noted that these profiles are very different from a Poiseuille type profile as the one observed for the flow of a single polymer. Velocity profiles of type (a) can be a source of coextrusion instability while profiles of type (b) are more conducive to stable flow. The shear rate and viscosity distributions within the thickness are drawn on Figure 11 for the velocity profile (a) of Figure 10.

The continuity of the shear stress at the interface between the layers results in discontinuities, which can be very large, in both shear rate and viscosity. This points out that the viscosity ratio of the two polymers, which is a key parameter in Newtonian cases, but which is also very widely used in practice in most coextrusion problems, loses all its meaning as soon as it is applied to non-Newtonian polymer fluids. The infinite viscosity calculated at the maximum velocity of Figure 10, therefore for a zero-shear rate, is an artefact of the power law viscosity model (Equation (10)). To account for the existence of a Newtonian plateau at low shear rate, it is necessary to use more sophisticated constitutive equations, as the generalized Carreau law [17]:(19)$$\eta =\eta_{0}{\left[1+{\left(\lambda \left|\dot{\gamma }\right|\right)}^{a}\right]}^{(m-1)/a}$$

$\eta_{0}$ is the viscosity at the Newtonian plateau. Parameters λ, *m* and *a* are fitted to experimental rheology curves. Accounting for this constitutive equation requires using numerical methods which will be presented in Section 2.3.

### 2.3. Transitory Phenomena at the Coextrusion Die Inlet

In Section 2.1 and Section 2.2, it has been supposed that the velocity profile was established immediately after the junction of the two monolayer streams. The question is: how long does it take to move from a discontinuous velocity profile at die inlet (Point A on Figure 12) to a continuous one (Point B)?

This requires a 2D modelling of the flow, initially proposed by [18,19]. An isothermal bilayer flow of two Carreau fluids, with a specific treatment of the interface conditions has been proposed by ([20,21]). One needs to solve the Stokes and continuity equations for each layer:(20)$$\nabla \cdot \left[2\eta_{i}\left(\dot{\bar{\gamma }}\right)\;\dot{\varepsilon }(u_{i})\right]=\nabla p_{i}\;\;\mathrm{with}:\;\dot{\varepsilon }(u_{i})=\frac{1}{2}(\nabla u_{i}+\nabla u_{i}{}^{t})$$
(21)$$\nabla \cdot u_{i}=0$$
with the standard conditions of continuity of velocities and stresses at the interface. Immiscibility of the two fluids at this interface is defined by u⋅n=0, where **n** is the vector normal to the interface. This last condition is equivalent to a condition of transport of the rheological constants in the whole area. Thus, for a Newtonian fluid, this condition is equivalent to u⋅∇η=0, where *η* is a discontinuous function, equal to $\eta_{1}$ in one layer and *η*_2_ in the other one. The determination of the free surface is obtained by solving the coupled Stokes equations and transport equation. The boundary conditions (Figure 12) are established Poiseuille velocity profiles in both feeding channels (Γ_1_ and Γ_2_) merging at point A, a zero velocity at the walls (Γ_3_) and an established flow at the exit Γ_4_ (for which the transverse velocity component vanishes). For a more complex rheological behavior based on the Carreau model (Equation (19)), it is sufficient to simply transport one rheological constant (e.g., *η*_0_, *λ*, *a* or *m*). An iterative fixed-point method is used to account for the nonlinear behavior.

Figure 13 presents an example of development of the interface. It can be faster or slower depending on the flow rate ratios, the viscous properties of the two fluids and the total flow rate. In the case presented here, for *Q*_2_*/Q*_1_ =1, a steady interface position is reached quite immediately, whereas for *Q*_2_/*Q*_1_ = 0.1, a distance of a few millimeters (for a gap of 2 mm) is sufficient to reach a steady state position. This justifies the stationary approaches presented in the previous paragraphs for die lands which length is of the order of several cm.

## 3. Miscellaneous Applications of Poiseuille Coextrusion Computations

### 3.1. Multilayer Non-Isothermal Coextrusion Flows

The bilayer coextrusion situation depicted in the preceding subsections is very far from usual coextrusion processes which involve multiple polymer layers extruded at different temperatures ([15,22]). Accounting for the heat transfer between the different layers and with the die walls induces a progressive modification of the temperature profile between the die walls. The viscosity will evolve accordingly in each coextruded layer along the flow direction and the hypothesis of a constant interface position, which has been used in the preceding paragraphs, is no more valid. Assuming that the different interfaces $\alpha_{i}$ vary slowly in the flow direction ($d\alpha_{i}/dx\leq 1$) the lubrication approximations may be applied. Using a power law behavior for each polymer, the preceding approach may be used assuming a constant velocity and shear stress at each interface. The flow rate in each layer $Q_{i}$ is deduced (Equations (17) and (18)), but it is now necessary to replace the constant pressure drop (Δp/L) by a local pressure drop (dp/dx) uniform through the die gap at each abscissa *x*, but varying along the flow direction. The temperature balance equation writes in each layer:(22)$$\rho_{i}c_{i}u_{i}\frac{\partial T_{i}}{\partial x}=k_{i}\frac{\partial^{2}T_{i}}{\partial z^{2}}+K_{i}{\left|\frac{du_{i}}{dz}\right|}^{m_{i}+1}$$

$T_{i}(x,z)$ is the temperature profile in layer *i.* The last term of Equation (22) is the viscous dissipation, $\rho_{i}c_{i}$ is the heat capacity, $k_{i}$ the heat conductivity, $K_{i}$ and $m_{i}$ the consistency and the power law index for each polymer *i.* Parameters $\rho_{i}c_{i},k_{i},m_{i}$ are supposed to be independent on temperature. The consistency $K_{i}$ will depend on temperature following an Arrhenius or a WLF equation ([2]). A controlled temperature, or a constant heat flux is imposed at the die walls. Heat flux continuity is imposed at each interface. It writes:(23)$${k_{i}\frac{\partial T_{i}}{\partial z}|}_{\alpha_{i}h}=k_{i+1}{\frac{\partial T_{i+1}}{\partial z}|}_{\alpha_{i}h}$$

$\alpha_{i}h$ is the interface position between layer *i* and (*i* + 1).

Knowing the *n* flow rate values $Q_{i}$, the problem consists in computing, at each abscissa *x*, starting from the lower die wall where a zero velocity is imposed, the (*n* − 1) interface locations $\alpha_{i}$, the pressure gradient (*dp*/*dx*), and the location of the maximum velocity *c.* To determine these (*n* + 1) unknowns, one writes *n* equations for the constant flow rates $Q_{i}$ in each layer and one imposes a zero velocity on the upper die wall. This problem is solved with a Newton-Raphson numerical method.

The velocity field $u_{i}$ is deduced in each layer (Equation (14) or (15)) and Equations (22) and (23) are solved through the die gap using a 1D finite difference method. The mean temperature is then computed in each layer to determine a new consistency value at the next abscissa (*x* + *dx*) using the Arrhenius or the WLF temperature dependence.

As an example, we consider the non-symmetrical case of a 6-layers coextrusion flow consisting of 6 different polymers (including two thin layers of adhesive) in the lip zone of a multichannel die (Figure 14). The geometry is rectangular (length of 80 mm, width of 300 mm and thickness of 2 mm). The total flow rate is 58 kg·h^−1^ and the final desired configuration is as follows (from bottom to top): 0.6 mm Polystyrene/0.1 mm adhesive/0.3 mm EVOH/0.1 mm adhesive/0.5 mm HDPE/0.4 mm LDPE. The die temperature is controlled at 200 °C and the initial temperatures of the various layers are largely different due to the melting process in each individual extrusion machine: 225 °C for Polystyrene, 190°C for the adhesive 1, 240 °C for EVOH, 200 °C for adhesive 2, 220 °C for HDPE and 170° C for LDPE.

Figure 15 presents the shape of the velocity profile near the die inlet. We note that the thickness distribution of the polymers in the flow is significantly different from the final distribution sought (Figure 14). It is observed, moreover, that the velocity profile maintains a fairly regular shape (e.g., relative to that of Figure 12, case a). A regular velocity profile similar to that of a single material flow is considered as a quality criterion to prevent defects that may occur at the interfaces due to too large velocity discontinuities between the various materials.

This velocity profile develops along the flow due to changes in temperature and viscosity of the various layers and, hence, it causes a change in the position of the interfaces. Figure 16 illustrates that the variation of the interface is particularly sensitive at the beginning of the flow and, then, the configuration remains almost constant. Figure 17 shows that the temperature profile, initially heterogeneous due to significant inlet temperatures differences between the different layers, becomes rapidly uniform and tends towards an equilibrium profile. These results demonstrate the low influence of the initial temperatures on the flow and are mainly explained by the very low thickness of each layer, which largely favor thermal conduction.

### 3.2. Investigation of Coextrusion Instabilities

Coextrusion instabilities (Figure 6) are different than classical extrusion instabilities (named sharkskin, melt fracture, stick-slip) which have been widely investigated in polymer ([23]) and food extrusion. These coextrusion instabilities develop at the interface between two coextruded polymers in flow situations where the outer extrudate surface remains smooth. They have been described by many authors ([7,21,22,24,25,26,27]). Using a very long flat die of rectangular cross section, Valette et al. [28] observed, on the solidified sample extracted from the die, the progressive development of a chevron shaped defect at the interface between both polymers as depicted in Figure 18. This defect was not visible at the die entrance, but it develops gradually towards the die exit.

It has been seen that the transitory phenomena induced by discontinuous kinematics and temperature fields at die inlet are very restricted in space and lead to stabilized interface positions, velocity and temperature fields in the major die land distance. These stationary velocity fields are a prerequisite for understanding the stability of the coextrusion process. Experiment presented Figure 18 pointed clearly out that the instability is wavelike and that its amplitude increases exponentially along the flow.

This suggests applying to all flow variables, for example the interface position between the polymers in a pressure flow (Figure 10), a disturbance of the following form:(24)$$\alpha (x,t)=\alpha_{s}+\alpha_{{}_{i}}\exp\left[i(kx-\omega t)\right]$$
where $\alpha_{S}$ is the position of the interface in stationary conditions, $\alpha_{i}$ is the initial disturbance which can be related to extrusion instabilities in the upstream flows (related for example to the periodic discharge between the metering zone of the extruder and the die inlet which is progressively amplified by the wear of the screw), *k* is a complex number (with a real part or wavenumber $k_{r}$ and an imaginary part $k_{i}$). The spatial period of the disturbance is $2\pi /k_{r}$ and (−$k_{i}$) is the spatial amplification rate; *ω* is a complex number for which the imaginary part, $\omega_{i}$, is the growth rate of the disturbance as a function of time (decrease rate, if negative) and $2\pi /\omega_{r}$ is the period ($\omega_{r}$ is the real part of *ω*). The first exponential term in Equation (24) expresses the amplification (or vanishing) of the initial disturbance, $\alpha_{i}$, in space and time; the second exponential expresses the shape of the disturbance.

(a)Asymptotic stability analysis

As the observed perturbations have large spatial amplitudes when compared to the flow gap, which means small $k_{r}$ values, it is interesting to look at the limit when *k* tends to zero: If *ω* has a positive imaginary part, an existing disturbance at *t* = 0 will grow with time. If, on the contrary, *ω* has a negative imaginary part, any disturbance will disappear with time. To capture flow instabilities, it is necessary to introduce viscoelastic constitutive equations which do not modify the stationary velocity field computed previously between parallel plates, but introduces time dependence. The first models developed in ([29,30,31]) used simplified viscoelastic constitutive equations (Maxwell, Oldroyd-B), far from the real behavior of polymer melts. Valette et al. ([32]) introduced a White-Metzner viscoelastic equation that better reflects the shear behavior of molten polymers to solve the coextrusion flow of Figure 10. Perturbations of the same type as in Equation (24), but with *k* = 0, were introduced into the coextrusion time-dependent equations (dynamic equilibrium, mass balance for each polymer, rheological behavior for each polymer, stress continuity and immiscibility between the fluids at the interface) for all variables (pressure, velocities, stresses, and interface position between the two fluids). These equations are then linearized about the stationary solution, assuming that the amplitude of the perturbation $\alpha_{i}$ is small. Writing the flow balance for each layer and imposing a zero velocity on the upper and lower walls leads to an eigenvalue problem, the solution of which is the value of ∂ω/∂k when *k* tends to 0.

Figure 19 reports stable (S) and unstable (U) flow conditions for the coextrusion geometry of Figure 8 for different polyethylene and polystyrene flow rates. Multiple transitions between stable and unstable flow conditions are observed. The agreement with experiment is qualitatively correct, but there is a point at low PE flow rate (point A) that is experimentally stable in an unstable calculated region and a point at high PE flow rate (point B) that is experimentally unstable in a stable calculated area.

(b)Convective Stability Analysis

In order to overcome the limitation of the asymptotic approach, it is necessary to solve the eigenvalue problem for each value of *k_r_*. This was achieved by [33] using a multimode viscoelastic behavior and by ([32,34]) using the White-Metzner equation previously used for the asymptotic analysis. The same approach, but now with non-zero *k* values, leads to more complex linearized equations with coupling of time and space scales. Figure 20 compares the experimental amplification rate of the interface perturbation reported in Figure 19, for different frequencies of the imposed perturbation (for example of the rotation velocity of one extrusion screw), with the theoretical amplification rate calculated by this convective stability analysis for the same flow conditions. It is observed, both experimentally and by calculations, that there is a critical frequency for which the amplification rate of the perturbation is a maximum. This means that some frequencies induced by the upstream machinery will be amplified and others not.

These convective stability calculations were then performed for each experimental condition shown in Figure 19. For each coextrusion condition, it leads to the disturbance frequency, but also to its spatial amplification rate, expressed in mm^−1^. An amplification rate of 0.01 mm^−1^ means that an instability of the order of 1 micron at die inlet will be amplified by a factor exp (0.4) to the end of a 40 mm length die; this means an amplitude of 1.49 micron that remains invisible to the naked eye. An amplification rate of 0.46 mm^−1^ means that the same initial instability will be sufficiently amplified to be visible at die exit. In fact, it would be amplified by a factor exp (18.4) at the die exit, which is out of the validity of the linear stability analysis and the calculations have no longer meaning.

This convective stability analysis points out that asymptotic unstable flow conditions, but with low amplification rates, may correspond to stable coextrusion experiments (point A for low PE flow rate in Figure 19). On the other hand, stable asymptotic coextrusion conditions (point B in Figure 19) may be convectively unstable. In the same way, the convective stability analysis points out that increasing the die length will amplify significantly the interface instability, as observed experimentally Figure 18, in contrast to what one could expect that increasing the length of the coextrusion die would relax an initial disturbance.

These approaches can be generalized to situations where there are more than two polymer layers, which correspond to the most coextrusion situations. However, they are difficult to apply to industrial geometries, for which there is no stationary analytical solution. This justifies using direct numerical simulation.

(c)Direct Numerical Modeling

The feasibility of direct numerical simulation of two-layer coextrusion flows was first demonstrated by ([35,36,37]). The main challenge is to capture the precise position of the interface between the two fluids as a function of time. As proposed in Section 2.3, the fluids immiscibility condition at the interface is changed in a transport equation of a characteristic function of each polymer, which is for example 0 in the first polymer and 1 in the second one. As a consequence, the rheological behavior of each fluid depends only on the characteristic function and the velocity and stress continuity conditions at the interface are automatically satisfied.

This approach has been applied to industrial conditions. The coextrusion of two polyesters in the final section of a coat-hanger die with a convergent gap and a constant width was considered. The two polyesters had different behaviors, one was Newtonian and the other one viscoelastic and shear-thinning, described by a multi-mode Oldroyd-B model ([38]). A small amplitude perturbation is imposed on one of the two inlet flows and the propagation of this disturbance along the flow is computed. Sometimes, this perturbation is simply transported as illustrated in Figure 21a. Under other coextrusion conditions, it is amplified (Figure 21b) and a change in the period of the perturbation is observed around the convergent part of the die.

## 4. Coextrusion in a Coat-Hanger Die

1D computations are not suited for analyzing the flow in feedblock die geometries (Figure 4). In that case, the major problem is to realize at die exit a uniform thickness distribution of the coextruded product, but also of each layer of this product. This would require 3D multilayer non-isothermal computations in a non-trivial geometry. Assuming that the die gap *h*(*x*,*y*) is small as compared to other dimensions of the die, it is possible to generalize the thin-layer approximations (called also Hele-Shaw approximations), already used for single polymer flow computations in several die or mold geometries ([2]), to multilayer flows. This reduces significantly the modeling complexity. The computations proposed by [39] for two-layer flows have been generalized by [40] to multilayer flows.

### 4.1. Bilayer 2D Coextrusion Flow

One considers the flow in the die geometry of Figure 22.

The velocity field is: $u_{i}(u_{i}(x,y,z),v_{i}(x,y,z),w_{i}(x,y,z))$ in each layer (*i* = 1, 2). As there is a sticking contact with the die walls (first layer with the bottom wall and second layer with the upper wall), the velocity gradient in the thickness is much higher than the velocity gradients in the in-plane direction (*x*, *y*). On the other hand, the vertical velocity components *w_i_* are much smaller than the mean velocity components in the horizontal plane (${\bar{u}}_{i},{\bar{v}}_{i}$).

The Stokes equations write in each layer:(25)$$\frac{\partial p_{i}}{\partial x}=\frac{\partial }{\partial z}[K_{i}{\dot{\bar{\gamma }}}_{i}{}^{m_{i}-1}\frac{\partial u_{i}}{\partial z}]$$
(26)$$\frac{\partial p_{i}}{\partial y}=\frac{\partial }{\partial z}[K_{i}{\dot{\bar{\gamma }}}_{i}{}^{m_{i}-1}\frac{\partial v_{i}}{\partial z}]$$
(27)$$\frac{\partial p_{i}}{\partial z}=0$$
(28)$$\frac{\partial u_{i}}{\partial x}+\frac{\partial v_{i}}{\partial y}+\frac{\partial w_{i}}{\partial z}=0$$
(29)$${\dot{\bar{\gamma }}}_{i}=\sqrt{{\left(\frac{\partial u_{i}}{\partial z}\right)}^{2}+{\left(\frac{\partial v_{i}}{\partial z}\right)}^{2}}$$
is the mean shear rate in layer *i* in each point of Ω.

Equation (27) means that the pressure is uniform through the thickness and so $p_{i}=p$. One first searches for a solution of Equations (25) and (26) as a product of a function of *z* and a function of (*x*, *y*) assuming, which is natural, that the velocity field is collinear with the pressure gradient:(30)$$\left\{ \begin{array}{l} u_{i}=\Phi (z){\left|\nabla p\right|}^{\delta_{i}}\frac{\partial p}{\partial x}\\ v_{i}=\Phi (z){\left|\nabla p\right|}^{\delta_{i}}\frac{\partial p}{\partial y} \end{array}\right.$$
(31)$$\left|\nabla p\right|=\sqrt{{\left(\frac{\partial p}{\partial x}\right)}^{2}+{\left(\frac{\partial p}{\partial y}\right)}^{2}}$$

As pointed out in [40], $\delta_{i}=\frac{1-m_{i}}{m_{i}}$.

Introducing the derivative of the velocity field (Equation (30)) as a function of *z* in Equation (29) leads to a new expression of the mean shear rate in layer *i.*
(32)$${\dot{\bar{\gamma }}}_{i}=\left|\frac{d\Phi (z)}{dz}\right|\nabla p^{{\frac{1}{m}}_{i}}$$

The velocity field (Equation (30)) is introduced in the Stokes Equations (25) and (26) with the mean shear rate value (Equation (32)). Computing then the pressure gradient (Equation (31)) leads to the following equation for Φ(z) in each layer:(33)$$\frac{d}{dz}(K_{i}{\left|\frac{d\Phi (z)}{dz}\right|}^{m_{i}-1}\frac{d\Phi (z)}{dz})=-1$$
with $K_{i}=K_{1}$ and $m_{i}=m_{1}$ for 0<z<αh(x,y); $K_{i}=K_{2}$ and $m_{i}=m_{2}$ for αh(x,y)<z<h(x,y). It is solved using the boundary conditions along the lower and upper die walls (Φ(0)=0 and Φ(h(x,y))=0) and a continuity condition at the interface z=αh(x,y)
(34)$$K_{1}{\left|\frac{d\Phi (z)}{dz}\right|}^{m_{1}-1}\frac{d\Phi (z)}{dz}=K_{2}{\left|\frac{d\Phi (z)}{dz}\right|}^{m_{2}-1}\frac{d\Phi (z)}{dz}$$

In order to calculate the velocity field (Equation (30)), one needs to determine the pressure field *p* (*x*, *y*). For that purpose, the continuity equation (Equation (28)) is integrated in each layer: It writes in the first layer:(35)$$\int_{0}^{\alpha h}\frac{\partial u}{\partial x}dz+\int_{0}^{\alpha h}\frac{\partial v}{\partial y}dz+w(x,y,\alpha h)=0$$

Using $\frac{\partial }{\partial x}\left(\int_{0}^{\alpha h}udz\right)=\int_{0}^{\alpha h}\frac{\partial u}{\partial x}dz+u(x,y,\alpha h)\frac{\partial (\alpha h)}{\partial x}$ and an equivalent expression for the *v* velocity component, (Equation (35)) writes:(36)$$\frac{\partial }{\partial x}\left(\int_{0}^{\alpha h}u(x,y,z)dz\right)+\frac{\partial }{\partial y}\left(\int_{0}^{\alpha h}v(x,y,z)dz\right)+w(x,y,\alpha h)-\frac{\partial (\alpha h)}{\partial x}u(x,y,\alpha h)-\frac{\partial (\alpha h)}{\partial y}v(x,y,\alpha h)=0$$

The non-miscibility at the interface between the two layers implies:$$w(x,y,\alpha h)=\frac{\partial \alpha h}{\partial x}u(x,y,\alpha h)+\frac{\partial \alpha h}{\partial y}v(x,y,\alpha h)$$
which simplifies Equation (36) in:(37)$$\frac{\partial }{\partial x}\left(\int_{0}^{\alpha h}u(x,y,z)dz\right)+\frac{\partial }{\partial y}\left(\int_{0}^{\alpha h}v(x,y,z)dz\right)=0$$

This equation expresses simply the flow balance equation in the first layer. Introducing the velocity components (Equation (30)) leads to:(38)$$\frac{\partial }{\partial x}\left(A_{1}(x,y){\left|\nabla p\right|}^{\delta_{1}}\frac{\partial p}{\partial x}\right)+\frac{\partial }{\partial y}\left(A_{1}(x,y){\left|\nabla p\right|}^{\delta_{1}}\frac{\partial p}{\partial y}\right)=0$$
(39)$$\mathrm{With}\;A_{1}(x,y)=\int_{0}^{\alpha h(x,y)}\Phi (z)dz$$

The same development in layer 2 leads to:(40)$$\frac{\partial }{\partial x}\left(A_{2}(x,y){\left|\nabla p\right|}^{\delta_{2}}\frac{\partial p}{\partial x}\right)+\frac{\partial }{\partial y}\left(A_{2}(x,y){\left|\nabla p\right|}^{\delta_{2}}\frac{\partial p}{\partial y}\right)=0$$
(41)$$\mathrm{with}:\;A_{2}(x,y)=\int_{\alpha h(x,y)}^{h(x,y)}\Phi (z)dz$$

In the case of two layers, we therefore have the two Equations (38) and (40). One of these two equations, Equation (38) for example, can be used to calculate the pressure assuming that the position of the interface α(x,y) is known which means that *A*_1_ can be calculated from Equation (39).

The boundary conditions are the followings (see Figure 22): *p* (*x*, *y*) = *P*_0_ on the entrance section $\Gamma_{e}$ and *p* (*x*, *y*) *=* 0 on the exit section $\Gamma_{s}$. Otherwise, a zero normal derivative of the pressure, which means a zero-flow rate, is imposed at the periphery of the die $\Gamma_{b}$: $\frac{dp}{dn}=0$, *n* is the normal to $\Gamma_{b}$. This zero-flow condition is less restrictive than a sticking contact along the periphery of the die and may induce some errors.

The pressure *p*, solution of Equation (38), minimizes the functional:(42)$$J(p)={\int_{\Omega }A(x,y)\left|\nabla p\right|}^{\delta }dxdy$$

Equation (42) is solved using a Finite Element method with the previously defined boundary conditions and with an iterative process when m≠1.

The second Equation (40) is then used to calculate the position of the interface. Indeed Equation (40) allows to calculate *A*_2_ if the pressure field *p* is known in the whole domain but the calculation strategy is different because the type of the equation is different. We define the components *U*_2_ and *V*_2_ of the mean velocity field within the layer thickness:(43)$$\left\{ \begin{array}{l} U_{2}={\left|\nabla p\right|}^{\delta_{2}}\frac{\partial p}{\partial x}\\ V_{2}={\left|\nabla p\right|}^{\delta_{2}}\frac{\partial p}{\partial y} \end{array}\right.$$

The trajectories of this vector field are the same as those described by the polymer even if the real travel velocity (which depends on the variable *z*) is not the same. Equation (40) is a transport equation for *A*_2_ along these trajectories. The resolution method is then different and several strategies can be used. It is a differential equation which can be solved step by step by following the trajectory since the value of *α* and therefore of *A*_2_ is known at the die inlet. This strategy is conceptually simple but technically complex. Another solution, technically simpler and now widely accepted, is to use finite element methods adapted to transport type equations such as discontinuous or off-center finite elements. Knowing *A*_2_, it is then possible to calculate *α* by Equation (41). A new pressure field is then computed by solving Equations (38) and (39) with the new interface position, and so on until convergence.

This method is generalized in the case of the coextrusion of n layers. The calculation of Φ(z) requires the introduction of n-1 continuity equations at interfaces of the type of Equation (34). Assuming that the interface between the first 2 layers is known makes it possible to have a first evaluation of the pressure using Equation (38) in the lower layer. We then determine the different $A_{i}(x,y)$ values and so of the different interface position $\alpha_{i}$ by solving equations of type (40) until convergence.

### 4.2. Comparison with Experiments

A three-layers (ABS/adhesive/PVDF) feedblock coat-hanger coextrusion die has been considered by Puissant et al. [41] (Figure 23). A power law rheology has been chosen for the three polymers: *K_ABS_* = 35,000 Pa·s^m^, $m_{ABS}$ = 0.29; *K_adhesive_* = 28,000 Pa·s^m^, $m_{adhesive}$ = 0.34; *K_PVDF_* = 19,600 Pa·s^m^, $m_{PVDF}$ = 0.44. The experimental polymer distribution at the feedblock location (point A on Figure 23) is indicated Figure 24. It needs to be respected at each iteration step. The measured entrance pressure is: *P*_0_ = 8 MPa.

Figure 25 compares the experimental interface positions at die exit to the computed ones. The final experimental distribution of the 3 polymers depends on their distribution at the coat-hanger die inlet, but the flow in the die exacerbates the initial thickness heterogeneities: for ABS the variation in thickness goes from 6% at the coat-hanger die entrance to nearly 20% at die exit; for PVDF the variations are even greater and we observe a virtual disappearance of PVDF on the edges of the final sheet. The consistency *K* of ABS is greater than that of PVDF and one might expect on the contrary an encapsulation of the ABS by the less viscous PVDF, but the shear-thinning index *m* of ABS is much lower than that of PVDF, which leads to local viscosity ratios that are out of all proportion to the consistency ratio. This example shows how a purely Newtonian analysis (constant viscosities) of the coextrusion of several polymers can lead to erroneous results. It is also observed that, if the adhesive keeps an approximately constant thickness over 90% of the final width of the die, it completely disappears at the periphery. This will cause delamination problems between the ABS and the PVDF and will require to remove the edges of the sheet.

The calculation qualitatively reflects these phenomena and, in particular, the disappearance of the binder at the die outlet. It is therefore an interesting tool to optimize the distribution of polymers in the coextrusion feedblock. It makes possible to produce a sheet in which the distribution of the thicknesses of the different polymers is as homogeneous as possible.

## 5. Encapsulation in Coextrusion

The most recent papers on the coextrusion process have been devoted to the encapsulation phenomenon, both experimentally and numerically.

### 5.1. Experiments

Several authors have studied this phenomenon in geometries of various sections (e.g., [5,24,42,43,44]). In Figure 26, Mauffrey ([[28])] shows the progressive encapsulation of the more viscous Polystyrene (in green) by the less viscous one (in white) in a very long channel (600 mm) of square cross-section (side 13.5 mm):

The parameters of the Carreau constitutive equation (Equation (19)) are listed in Table 1 for the two polystyrenes

Even if PS B (white) is only slightly less viscous than PS A (green), PS B encapsulates rapidly PS A till reaching the bottom corner (*z* = 120 mm). The polymer distribution remains then blocked around the corners for a long distance (120 mm < *z* < 240 mm) and, simultaneously, the top of the green polymer rises progressively towards the upper wall of the square section. Afterwards, recirculating cells of the green polymer are observed around the bottom corners. Finally, the encapsulation phenomenon progresses along the bottom side of the square section and is nearly completed at the die outlet (*z* = 580 mm). A small amount of green polymer remains blocked in the corners and then progressively spreads along the periphery of the square die. The dynamics of encapsulation is highlighted by measuring the perimeter fraction wetted by the green polymer as a function of *z* (Figure 27). A rapid encapsulation is observed towards the bottom corner followed by a steady position for around 100 mm. Then encapsulation restarts, but less rapid than in the first sequence.

Dooley et al. ([46]) showed that encapsulation can even occur when coextruding two slices of the same polymer but colored with different pigments (Figure 28).

In most industrial coextrusion situations, the final common die land is much shorter and the shape factor of the die cross section is small which means that the encapsulation will remain limited. However, an important coextrusion flow length with a higher shape factor exists in the distribution channel of the coat-hanger die after the feedblock (Figure 23).

### 5.2. Modeling

The physical phenomena governing the encapsulation phenomenon is paradoxical. As recalled by [47], a simple and intuitive interpretation is given by the so-called Principle of Minimization of Viscous Dissipation as formulated by [48]. Despite this intuitive explanation, the reality is much more complex.

Preliminary simple 2D computations of the Poiseuille flow of two purely viscous fluids for a given interface and given flow rates show that the pressure drop, and hence the viscous dissipation, decrease significantly if the less viscous fluid wets totally the die wall and hence lubricates the flow as underlined experimentally in [49]. Once the more viscous fluid is encapsulated by the less viscous one, it is again possible to continue minimizing viscous dissipation by optimizing the shape of the interface, even if benefit is much less important as soon as complete wetting is reached.

Numerical difficulties increase significantly when one tries to understand the complete phenomenon from the junction of the two fluids to the end of the die. A realistic modeling is necessarily 3D and one has to describe precisely the interface between the two components. These both peculiarities increase significantly time computation and numerical storage. Furthermore, it is then necessary to cope with the evolution of contact line between the two fluids and the wall along the die length. That computation of the interface generally uses a transport equation, for example the phase field equation, involving the velocity field which is zero at the wall. It does not mean that the interface at the wall is not able to evolve but that the equation used to simulate this evolution degenerates at the wall. If zero-velocity at the wall is a convenient boundary condition for most molten polymer flows to predict pressure/flow-rate relationships, it is unable to predict a finite residence time in the die. This points out that this boundary condition does not account for the complex displacement of macromolecules along the solid die surface ([50]). Furthermore, if the coextruded polymers are compatible (this is the case with two Polystyrenes Figure 26) these displacements at the micro or nano-level will induce a partial mixing between both polymers and this is visible on Figure 26, where the green and white Polystyrenes are locally mixed in the bottom corner.

At a macroscopic level, as observed in [46] for a single polymer flowing in a die of square cross section, recirculating cells are observed in a plane perpendicular to the flow direction. Such recirculating cells are observed Figure 26 when the contact line between both polymers touches the bottom corners of the square die. It can induce a local stratification of the flow in several thin layers (Figure 26, *z* = 440 mm). If the precise description of this complex topology is numerically difficult it is also of weak interest for understanding encapsulation problems occurring in polymer processing.

The first 3D purely viscous coextrusion computations ([51]) did not predict accurately the experimental displacement of the free surface. Teixeira-Pires ([44]) and Mauffrey ([5]) introduced in a purely viscous model a slip velocity around the triple point A (Figure 26), which is a function of the local viscosity ratio. Hesheng et al. ([52]) introduced a Navier slip law. Dooley et al. ([53]) introduced viscoelastic constitutive equations with a non-zero second normal stress difference in simple shear. They reveal the existence of several recirculating cells in the plane perpendicular to the flow direction. This has been used for coextrusion flows in ([54]) using a decoupled velocity and stress field computation and a high order polynomial development for extrapolating the contact line (point A, Figure 26) and by Anderson et al. ([45]) using a mapping technique to track the interface. Yue et al. ([55]) and Borzacchiello et al. ([47]) assumed a diffuse interface. In his PhD ([56]) he compared successfully the computation results with Teixera-Pires experiments (Figure 29).

These last authors introduced a phase field method with a scalar function representing the volume fraction of each phase to describe the interface. A second order finite volume method is used to solve the whole system of equations: mass and stress balances, viscoelastic Giesekus constitutive equation (which presents a second non-zero normal stress difference in simple shear) and phase field equation. It allows to predict a significant change of wetting and a displacement of the triple line despite the zero velocity at the wall. The numerical results are consistent with Teixeira-Pires experiments. When using purely Newtonian behaviours encapsulation was not found even in a wide range of viscosity ratios (between 1 and 0.05). However, the difficulty is then to measure this second normal stress difference and to fit parameters of constitutive equations to these measured values.

## 6. Conclusions

Coextrusion modeling makes it possible to understand the main features of the process. In the terminal land of multi-channel coextrusion dies encountered in film blowing, blow molding of hollow plastic bodies, manufacture of tube, cable coating and some sheets, the interface position between the different polymer layers is established within a few millimeters and the temperature is generally rapidly homogenized and controlled by the regulation temperature of the die. Encapsulation is obviously not present in axisymmetric flow geometries and limited in sheet extrusion as the sheet shape factor (*h/W*) is very small. As a consequence, the velocity and temperature profiles, the interface positions are established in the major part of the die and can be calculated easily with different rheology constitutive equations. The Newtonian solution is interesting as an academic exercise, but it is source of misunderstanding as the viscosity ratio is changing within the die gap. It is possible to obtain realistic analytical solutions using power law behaviors. Viscoelastic solutions may also be obtained which makes it possible to study time dependent solutions. Convective stability analysis shows that, depending on the flow rate of each polymer, initial perturbations initiated in the upstream extrusion machinery will be amplified or will vanish. It will also reveal the existence of critical frequencies for which the amplification is maximum. Direct numerical simulations may be performed in more complex geometries where there is no stationary analytical solution but the computation time step needs to be carefully adjusted, and each flow situation will require a new computation. These results are interesting on a processing point of view. As, at a first glance, one may believe that increasing the die length would damp down some upstream instabilities, it is necessary to limit the die land length in order to avoid spatial development of convective coextrusion instabilities. On the other hand, it is necessary to identify frequencies developing the most catastrophic convective instabilities. These frequencies may be induced by progressive wear of a feeding extruder.

In more complex flow geometries, as the one encountered in coat-hanger dies, the models also make it possible to optimize the design of the feedblock (the position and shape of the slats (see Pinsolle [1]) to obtain a uniform distribution of the different layers at the die outlet. A uniform distribution of the thicknesses of the different layers at the feedblock will not necessarily induce an optimal distribution in the final coextruded sheet, but the shape of the lamellae should be machinable and allow easy adjustment. The calculation can make it possible, by successive iterations, to propose an optimized feedblock geometry thus producing a sheet whose total thickness and the thickness distribution of the various constituent polymers will be uniform. Note that square or rectangular shapes with low slenderness can cause recirculation phenomena in the angles which interfere with the subsequent flow in the die and exacerbate the encapsulation phenomena.

The encapsulation phenomenon is a critical point in this coat-hanger geometry because, even if the shape factor of the final sheet is very low, the distribution channel upstream has an important shape factor (Figure 25). Encapsulation will develop in this area and then be convected in the final part of the die resulting in some polymer missing at the periphery of the sheet. The existing models for encapsulation are not totally convincing even if some physical features have been accounted for: viscoelasticity for the development of transverse recirculating cells, slip at the micro-level near the triple point, possible mixing of the two polymers in an intermediate layer. It is certainly a research area which deserves to be developed.

## Figures and Tables

**Figure 1 polymers-14-01309-f001:**
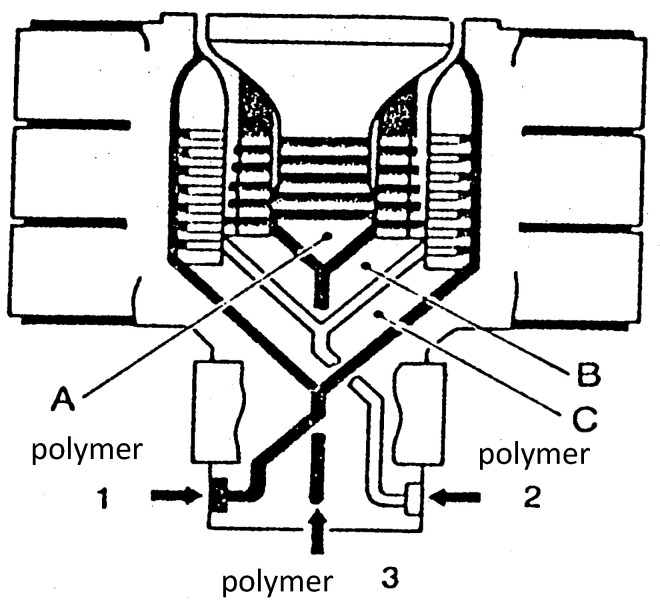
Example of a multilayer film blowing die geometry (Reprinted from Ref. [2]).

**Figure 2 polymers-14-01309-f002:**
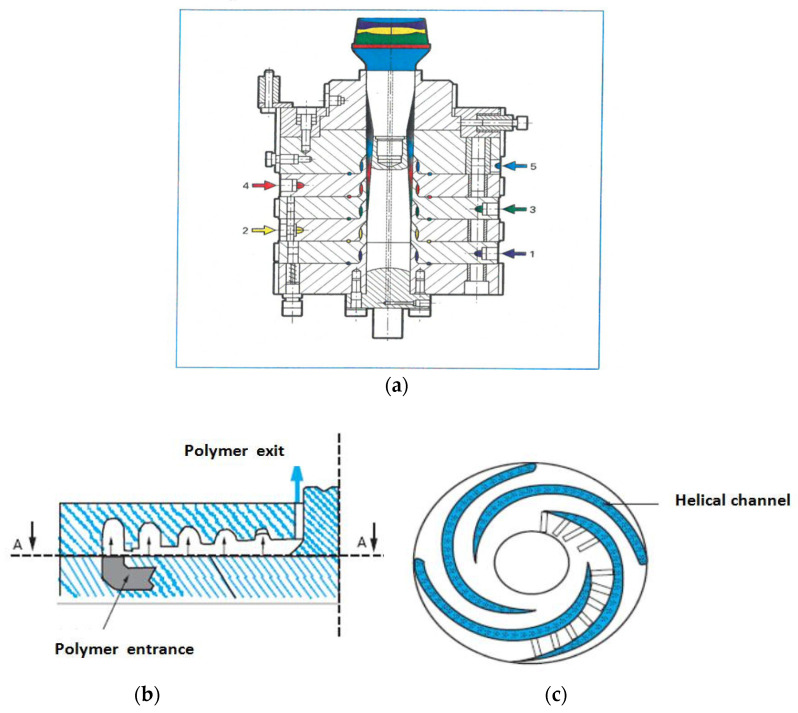
Example of a pancake die geometry; (**a**) stack of five pancake dies, (**b**) cross-section, (**c**) section AA (Reprinted from Ref. [2]).

**Figure 3 polymers-14-01309-f003:**
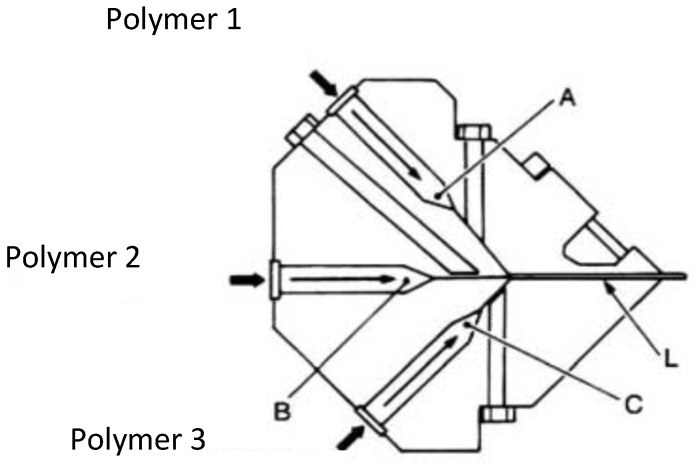
Multi-channel coextrusion die for sheets (Reprinted from Ref. [2]).

**Figure 4 polymers-14-01309-f004:**
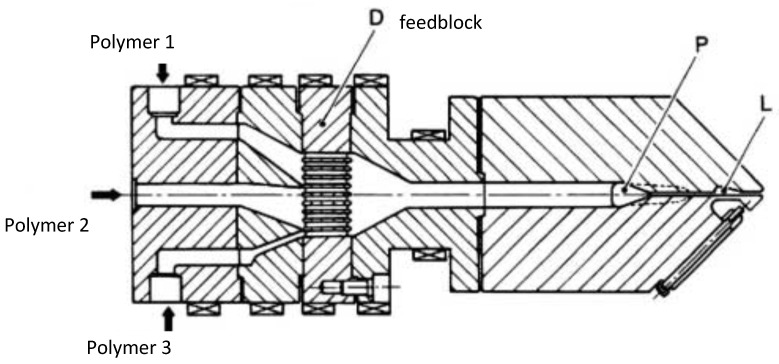
Feedblock die (Reprinted from Ref. [2]).

**Figure 5 polymers-14-01309-f005:**
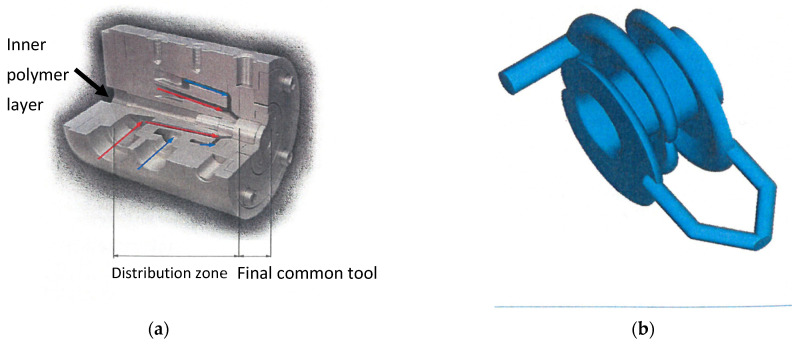
Geometry of the tube coextrusion die: (**a**) sectional view of the die; (**b**) unrolled “coat-hangers”. Reprinted with permission from Ref. [4]. Copyright 2009 Éditions Techniques de l’Ingénieur.

**Figure 6 polymers-14-01309-f006:**
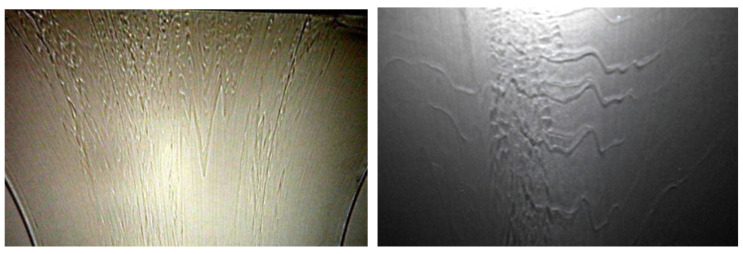
Typical defects observed in a coextrusion process (the flow is from top to bottom) (Reprinted from Ref. [2]).

**Figure 7 polymers-14-01309-f007:**
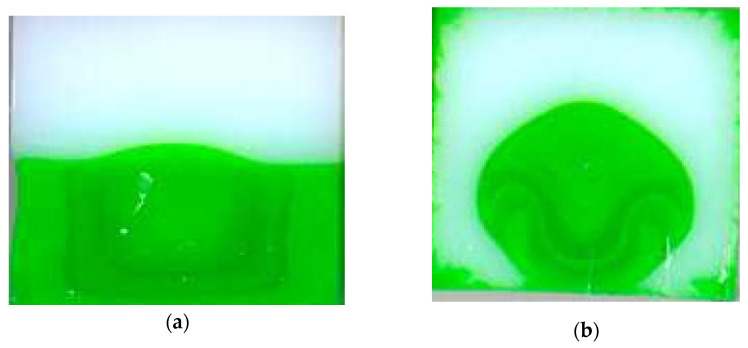
Typical encapsulation for the coextrusion flow of two different polystyrenes (white colored on the top and green colored on the bottom): (**a**) die entrance; (**b**) die exit (Photography reprinted from Ref. [5]).

**Figure 8 polymers-14-01309-f008:**
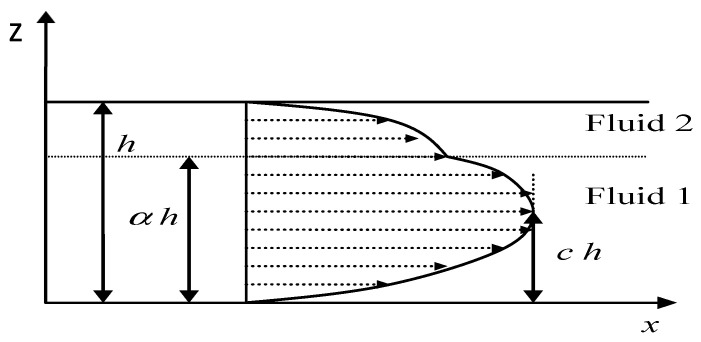
Two-layer flow between parallel plates.

**Figure 9 polymers-14-01309-f009:**
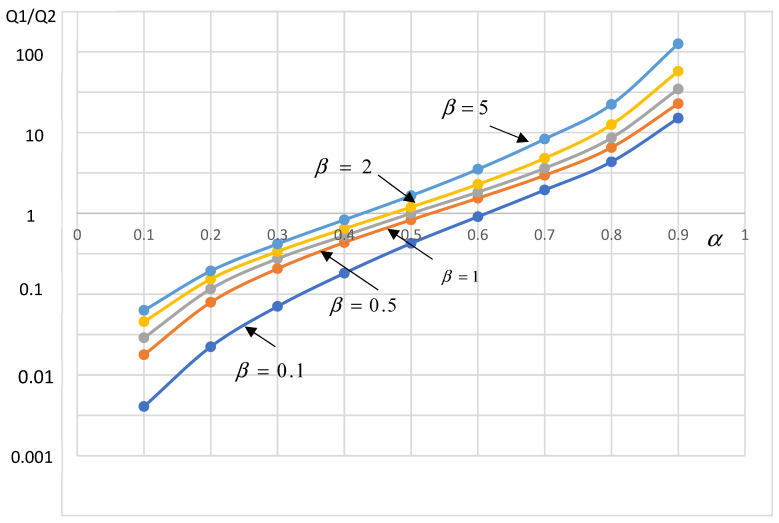
Interface position α as a function of the flow rate ratio for different values of the viscosity ratio β.

**Figure 10 polymers-14-01309-f010:**
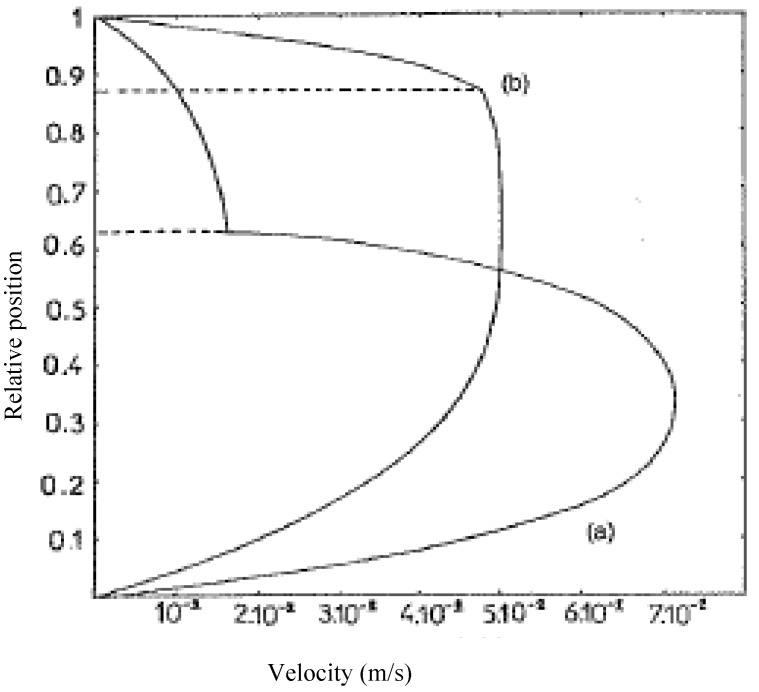
Velocity profiles for the flow of two fluids. Fluid 1: *m*_1_ = 0.5; (a) K1 = 10^3^ Pa·s^n^; (b) K1 = 10^4^ Pa·s^n^. Fluid 2: *m*_2_ = 0.3; K2 = 10^2^ Pa·s^n^ (Reprinted from Ref. [16]).

**Figure 11 polymers-14-01309-f011:**
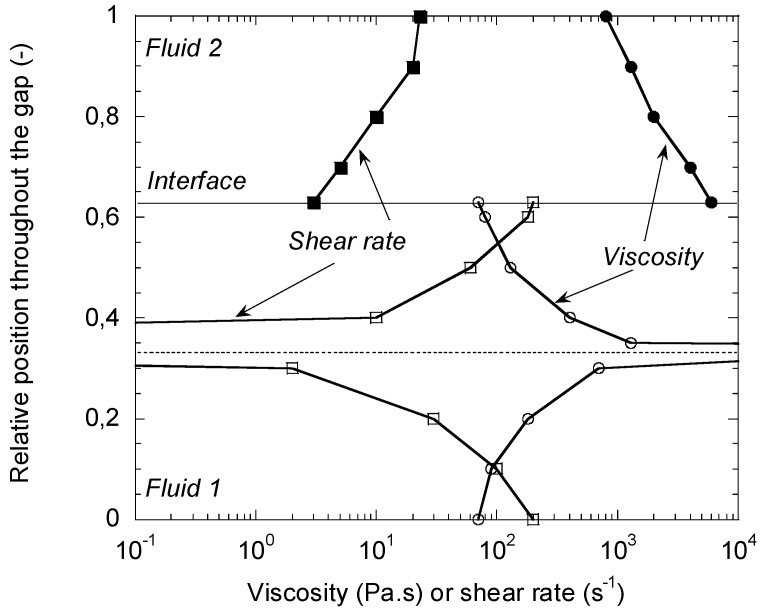
Shear rate (■, □) and viscosity (●, ○) through the gap for the velocity profile (a) of Figure 10 (Reprinted from Ref. [2]).

**Figure 12 polymers-14-01309-f012:**
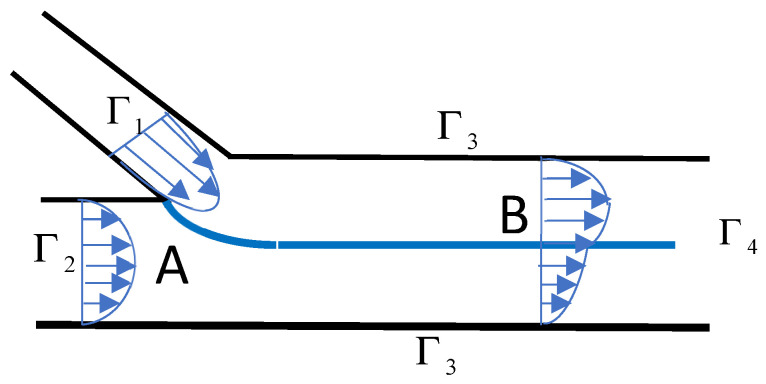
Convergence of two polymer layers at the entrance of a bilayer coextrusion die.

**Figure 13 polymers-14-01309-f013:**
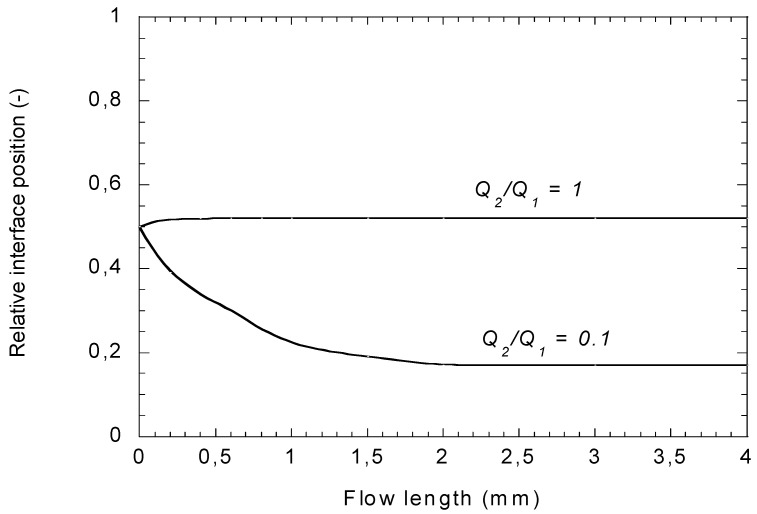
Variation of the interface position in the case of a bilayer flow of two Carreau fluids for two flow rate ratios. The flow rate *Q*_1_ = 2 × 10^−4^ m^3^/s for a die width of 1 m.

**Figure 14 polymers-14-01309-f014:**
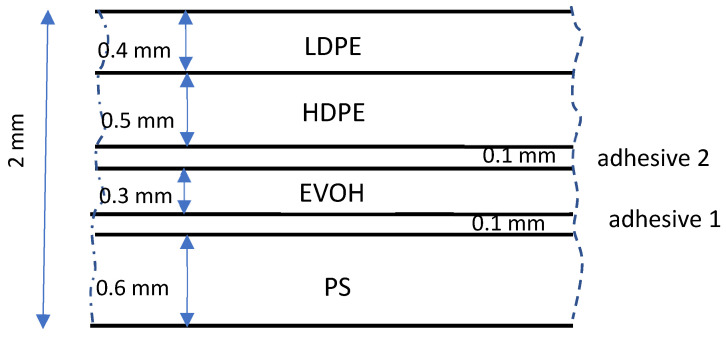
Typical multilayer coextruded product.

**Figure 15 polymers-14-01309-f015:**
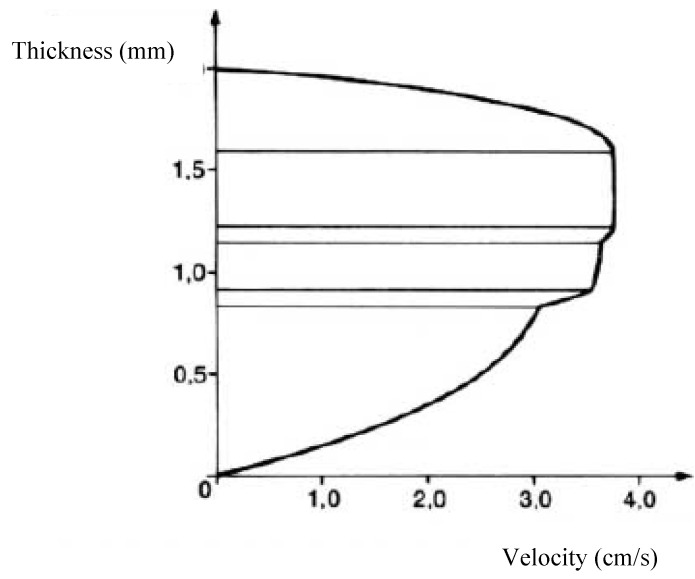
Velocity profile in a multilayer flow (Reprinted from Ref. [15]).

**Figure 16 polymers-14-01309-f016:**
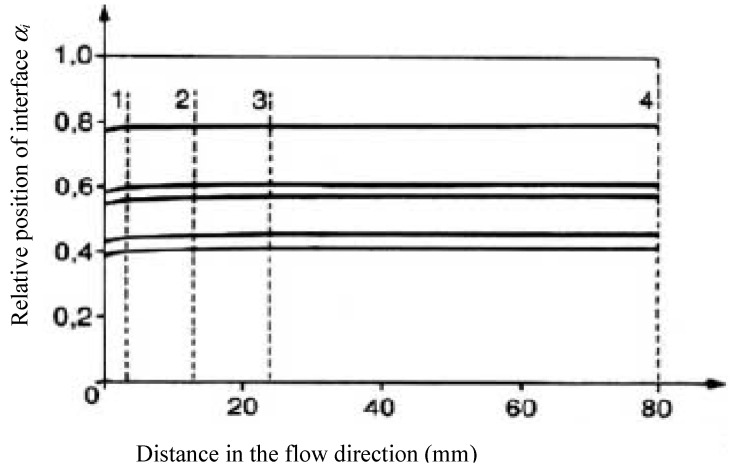
Variation of the interface positions along the coextrusion flow (Reprinted from Ref. [15]).

**Figure 17 polymers-14-01309-f017:**
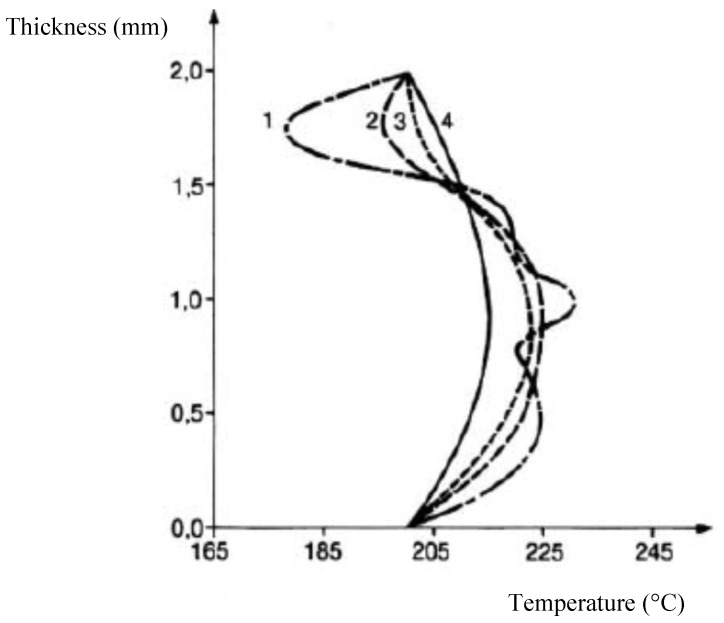
Variations of the temperature profile along the flow (the positions 1 to 4 are shown in Figure 16) (Reprinted from Ref. [15]).

**Figure 18 polymers-14-01309-f018:**
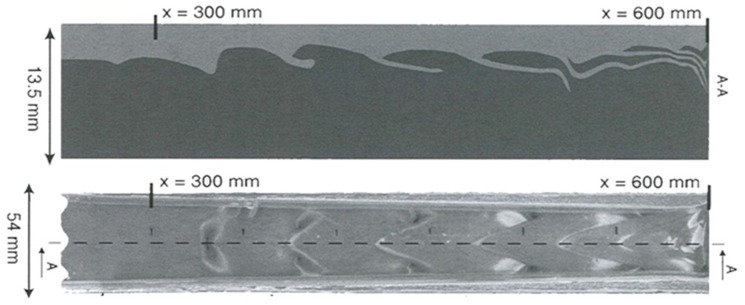
Coextrusion of a polyethylene (**bottom**) and a polystyrene (**top**). View of the second half of the extracted sample. Bottom photo: top view; upper photo: side view. The flow was from left to right (Reprinted from Ref. [28]).

**Figure 19 polymers-14-01309-f019:**
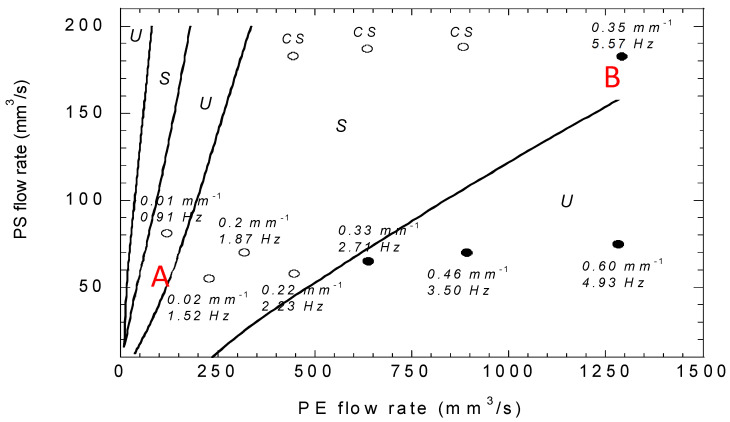
Stability of the coextrusion pressure flow of a polystyrene and a polyethylene at 200 °C; open symbols correspond to experimentally stable conditions and filled symbols are for experimental unstable conditions. Asymptotic stability calculations with a White-Metzner viscoelastic constitutive equation delimit the stable (S) and unstable (U) zones (Adapted from Ref. [32]).

**Figure 20 polymers-14-01309-f020:**
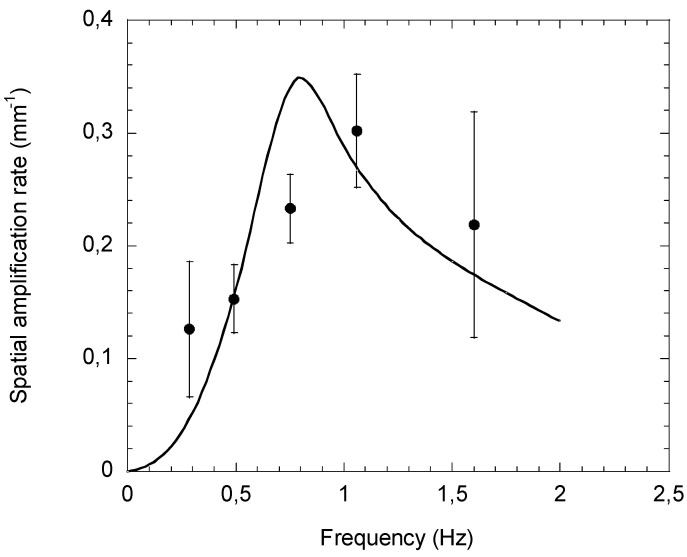
Spatial amplification rate as a function of the imposed perturbation frequency of the rotation velocity of one extrusion screw, (●) experimental points (—) convective stability calculations (Reprinted from Ref. [35]).

**Figure 21 polymers-14-01309-f021:**
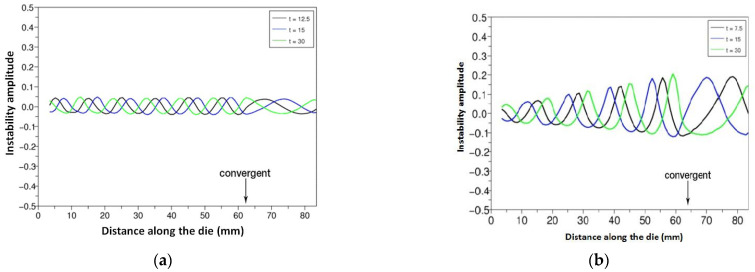
Propagation of an interface perturbation along the die geometry. (**a**) under stable conditions; (**b**) under unstable conditions (Reprinted from Ref. [38]).

**Figure 22 polymers-14-01309-f022:**
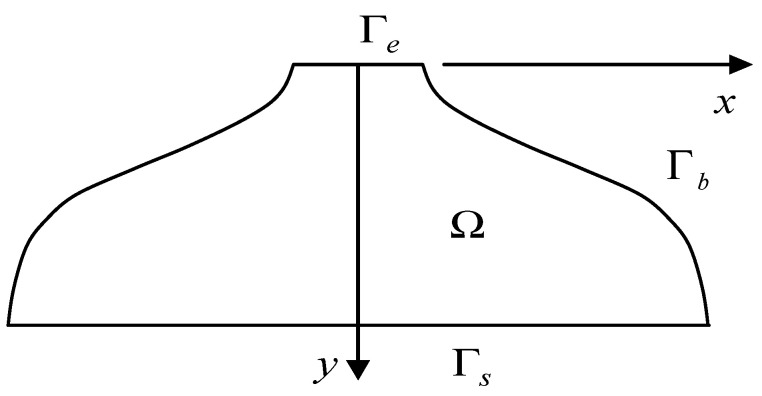
Scheme of the bilayer coextrusion die geometry.

**Figure 23 polymers-14-01309-f023:**
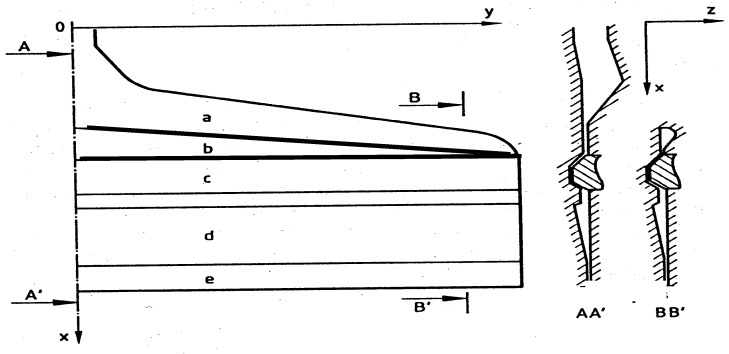
Coat-hanger die geometry. Only half of the geometry is represented (Reprinted from Ref. [2]).

**Figure 24 polymers-14-01309-f024:**
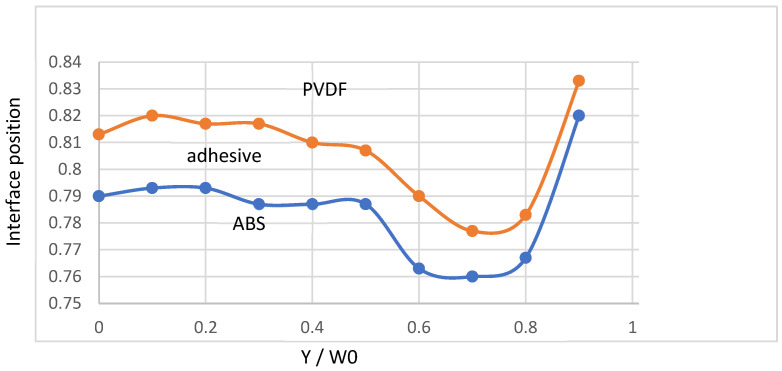
Polymer distribution at the coat-hanger coextrusion die inlet: The interface positions $\alpha_{1}$ and $\alpha_{2}$ are restricted to 0.75<α<0.84 in order to visualize the very thin adhesive layer (Adapted from Ref. [14]).

**Figure 25 polymers-14-01309-f025:**
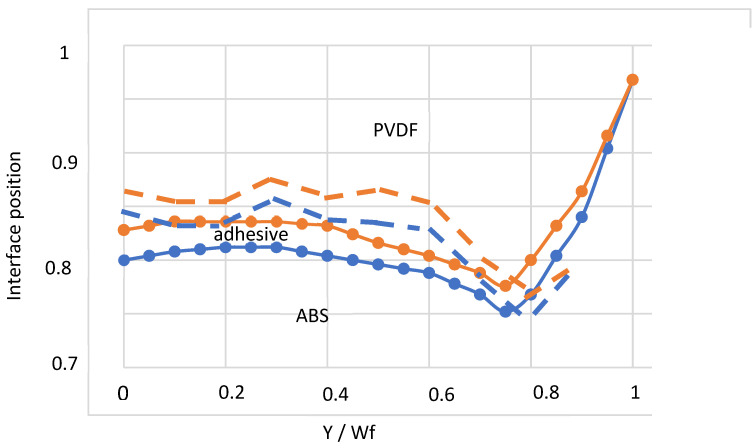
Polymer distribution at die exit: (**- - -**) computation: (•−•) experience. The interface positions ($\alpha_{1},\alpha_{2}$) are restricted to 0.7<α<1 in order to visualize the very thin adhesive layer (Adapted from Ref. [14]).

**Figure 26 polymers-14-01309-f026:**
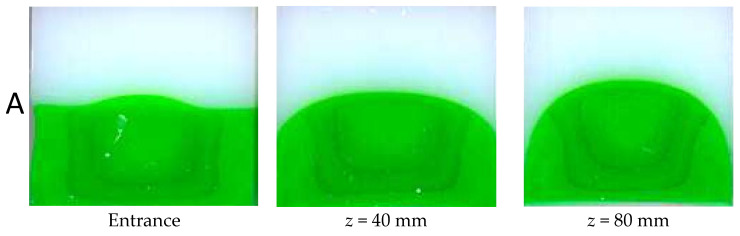
Coextrusion of two polystyrenes (white on the top, green on the bottom) in a long-slit die of square cross-section: successive cuts along the die length (photography reprinted from Ref. [45].

**Figure 27 polymers-14-01309-f027:**
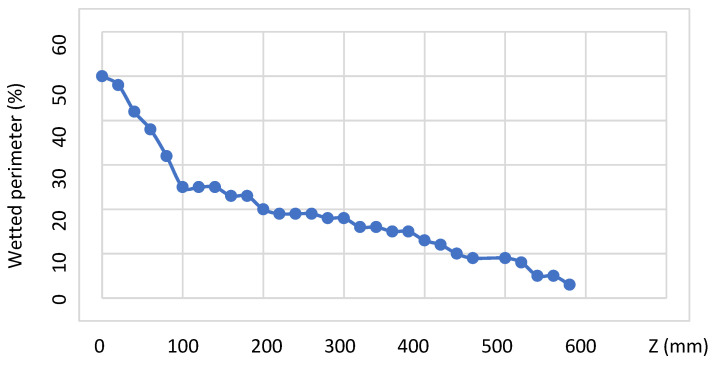
Wetted perimeter of the more viscous (green) polymer, as a function of the flow distance in the slit die of square cross-section (Adapted from Ref. [5]).

**Figure 28 polymers-14-01309-f028:**
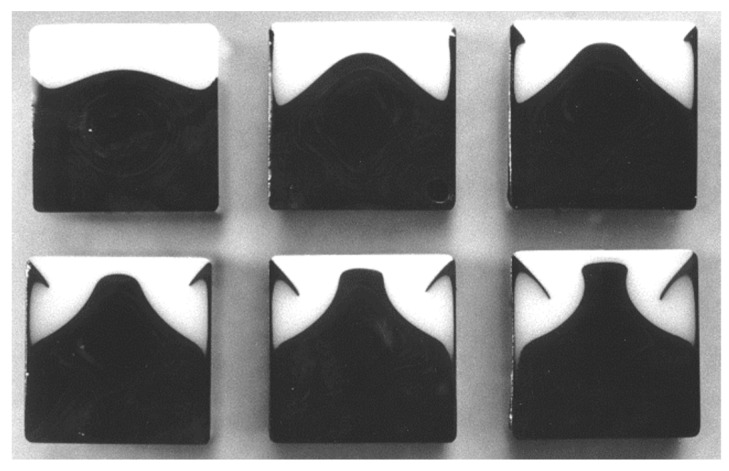
Encapsulation phenomenon observed in the coextrusion flow of two layers of the same Polystyrene with different colors (Reprinted from Refs. [11,45]).

**Figure 29 polymers-14-01309-f029:**
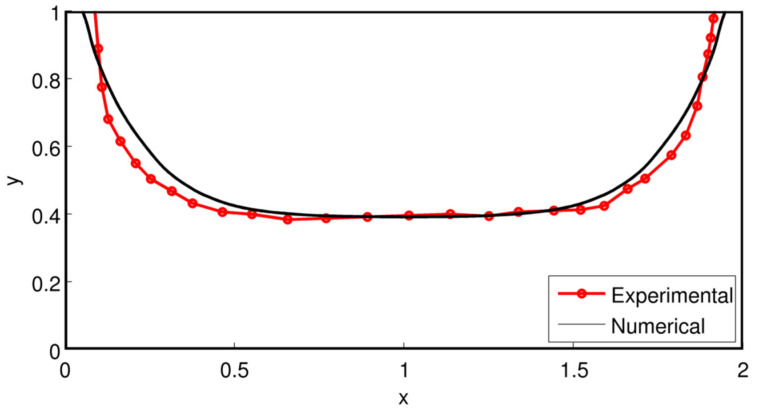
Coextrusion of two polystyrene in a slit die of rectangular cross-section: Comparison between the experimental data of Teixeira-Pires (Reprinted from [44]) and the numerical computations of Borzacchiello (Reprinted from [56]).

**Table 1 polymers-14-01309-t001:** Rhelogical parameters of the two polystyrenes (Generalized Carreau law).

	PS A	PS B
$\eta_{0}$ (Pa.s)	6750	6050
λ (s)	0.45	0.34
m	0.43	0.39
a	0.87	0.74

## Data Availability

Most of the data are available in the referenced papers or PhD.

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
