# Peer review of "Investigation of the Polymer Coextrusion Process: A Review"

_polymers, 2022, doi:10.3390/polym14071309_

Round 1

Reviewer 1 Report

Dear Authors,

Generally, this is a very interesting manuscript. I also believe that the authors have sufficiently described the phenomena related to co-extrusion of polymers. The scientific discussion is also well conducted, but sometimes there are no references to literature. To improve the quality of this article, a few more minor adjustments need to be made.

The following detailed comments:

Chapter 1 Comment: Explain what you mean by "use a recycled polymer of lesser quality".

You should also add the purpose of your work. Answer the question why you want to publish your article. What are these descriptions for. The subject of extrusion is still top of the line. You shouldn't have any major problems with that.

Comment on the chapter. 2.2: I think you should better explain what could be the source of coextrusion instability.

You should also refer to literature. Extruders are machines with continuous operation, therefore, when there are certain changes in operating parameters (e.g. change of rotational speed of screws), it may translate into pressure changes. Along with this parameter, the vibration frequency of the extruder also changes. The stability of this parameter determines the quality of the final products obtained. Interesting research in this field (with the use of wavelet analysis) was carried out by examining the food extruder. Check out the article "The use of wavelet analysis to assess the degree of wear of working elements of food extruders" below. The changes in the frequency spectrum were used here to assess the degree of wear of the working elements (screw and cylinder). An important parameter for maintaining the constant quality of products. For example, when the extrusion pressure starts to drop, you can increase the rotational speed a little - then the frequency will change a little ... .. Here you can also find something about the frequency in the extrusion process "Effects of extrusion frequency on the quality characteristics of Ddukgukdduk"

At the end of the work, it is also worth mentioning that the theoretical models refer to constant parameters (extruder without signs of wear). During operation, the extruder screws and the cylinder are subject to gradual wear all the time. When planning the technological process, this should be taken into account.

Comment on the chapter. 3.2: Here you can also refer to the literature in which the influence of the nozzle length on the process stability is described. In general, this also translates into the quality of the materials obtained.

Author Response

Please see the attachment, thanks for the review.

Reviewer 2 Report

This manuscript is a review of polymer coextrusion process, especially on the author’s group researches. As mentioned in the introductory section, coextrusion processes are now one of important solution for plastic products related to environmental problems of polymer recycle, and it is timely to review the practical subject of modeling the polymer coextrusion process. While the most part of this manuscript has been well reviewed based on historical developments, there are several attentions for publishing. I recommend to rewritten and add the explanation for understanding the meaning of figure as follows.

  1. Citation of figures should be checked even the authors group research. Figures 1, 3, 4, 6,7,11,15, 16,17,19,25,26,27,28 are seemed to be cited from other published papers.
  2. It is needed to explain the figures for understanding the contents. For examples, parts A, B, and C in figure 1, parts A, B, C, and L in figure 2, should be explain.
  3. In figure 7, change capital word for the fist paragraph. Please explain the color region and cross-sectional position.
  4. In page 7, 3 lines from top, alpha needs to be explained. The lines of equation number (2) and (3) are misaligned.
  5. In figure 9, the position of β is misaligned.
  6. In equation (20), unify the symbols used for the velocity gradient tensor.
  7. In page 12, there is blank in 15 line from the bottom.
  8. In figure 13, the flow condition and the angle of convergence are needed.
  9. In page 17, 2π/ωr is twice written.
  10. In page 28, 3D viscoelastic two layers simulation for encapsulation should be cited for the explanation of phenomena. For example, M. Takase, et al., Rheol. Acta 37, 1998, 624-634.
  11. References should be checked because there are several mistakes in omitting. For example, Trans. So.c. Rheo may be Trans. Soc. Rheol., J. Non-Newtonian Fluid Mech. In addition, there is a space in the line of Fortin, Ganpule, Puissant’ papers.

Author Response

(The authors gave the same response as above.)
